# Contributing to SDG 3 through Support for Trainer Autonomy: Relationship with Motivation, Basic Psychological Needs and the Intention to Be Physically Active

**DOI:** 10.3390/ijerph191912327

**Published:** 2022-09-28

**Authors:** Diego Andrés Heredia-León, David Manzano-Sánchez, Alberto Gómez-Mármol, Alfonso Valero-Valenzuela

**Affiliations:** 1Faculty of Sport Sciences, University of Murcia, 30100 Murcia, Spain; 2Academic Unit of Education, Catholic University of Cuenca, Cuenca 010105, Ecuador; 3Faculty of Education, University of Murcia, 30100 Murcia, Spain

**Keywords:** support for autonomy, self-determination theory, motivation, physical activity, teacher’s interpersonal style

## Abstract

The objective of the study was to identify the profiles of support for autonomy perceived by athletes and compare them with motivation, basic psychological needs, and the intention to be physically active. A cross-sectional descriptive study was carried out with 280 athletes (M = 15.28; SD = 1.71). The results reveal the existence of two profiles using the perception of support for autonomy as a grouping variable. A profile of “High perception of support for autonomy” (group 1, n = 190, M = 4.38, SD = 0.32) and a profile of “Low support for autonomy” (group 2, n = 90, M = 3.40, SD = 0.86). The data show statistically significant differences in favour of group 1 with higher values in the Self-Determination Index (*p* = 0.009) and three basic psychological needs measured with the Index of Psychological Mediators (*p* = 0.000), but no significant differences in intention to be physically active were found. In conclusion, the importance of the support of autonomy granted by coaches is demonstrated, thus favouring more adaptive motivational climates and the greater satisfaction of basic psychological needs. The intention to be physically active was not found in federated athletes. New studies are required to clarify this fact and improve the contribution of sustainable development to the social dimension.

## 1. Introduction

On the sports field, one of the determining aspects that influences people’s commitment is motivation [1]. Of the several perspectives that have been adopted in studies, the one that has stood out the most is the SDT self-determination theory [2,3,4]. This theory, established by Deci and Ryan [2,5,6] is a construct that focuses on distinguishing motivation at different levels of self-determination [2]. Vallerand [7] reveals that greater autonomy facilitates the existence of more self-determined motivation. Motivation is conditioned by the interpersonal style of the coaches [8], in such a way that the behaviour of the coaches is one of the most important aspects during the practice of sport [9]. For their part, Gillet, Vallerand, Amoura, and Baldes [10] state that researchers who use SDT are mainly interested in the impact of two interpersonal styles, the style that supports autonomy. On the one hand, this allows athletes to participate in the decision-making process and acknowledge their feelings and, on the other hand, encourages a controlling style characterized by a highly directive interaction on the part of the coach [9]. In this context, several investigations [10,11,12,13,14] have analysed the athletes’ perception of their coaches’ interpersonal styles, reflecting that controlling behaviours minimize self-determined motivation, while the support for autonomy promotes a level of motivation. The SDT also differentiates between autonomous motivation and controlled motivation, considering that the environment is an important factor for subjects to oscillate in a more or less autonomous or controlled way [3].

The SDT also highlights the importance of the motivational dimensions that determine how one adapts to and carries out a physical activity [15] in such a way that it implies the existence of basic psychological needs [3]—specifically the need for competence (e.g., focused on mastering abilities), autonomy (e.g., oriented to the use of good actions), and the need for a relationship (e.g., connecting with and giving affection to others)—which play an important role when it comes to the level of satisfaction [3]. Thus, in environments where autonomy is supported, the satisfaction of basic psychological needs will develop, while in controlling contexts, frustration and discomfort occur with these needs [4,16].

As for the support for autonomy, the basic psychological needs, when developed, promote optimal functioning and the progress of motivation, while their frustration hinders it [13]. In this way, the SDT suggests that when psychological needs are satisfied, the self-determined motivation of athletes is favoured, while their frustration promotes non-self-determined motivation [4,17]. In recent years, several investigations carried out in the sports environment have studied the predictive power of the satisfaction of needs on self-determined motivation [18,19,20,21,22], coinciding with a positive relationship between these variables.

The relationship between motivation and the intention to be physically active has been studied based on motivational profiles. The self-determined profiles are those that are related to the intention of future sporting practice [23,24,25,26], where it is demonstrated that the more self-determined profiles are related to a greater intention to be physically active, as well as a better physical self-concept, which can work as a facilitator of physical activity [27]. In this way, most scientific studies confirm that the perception of support for autonomy, the satisfaction of basic psychological needs, and motivation are predictive variables of sports practice [1]. Within the framework of the United Nations plans that focus on 17SDG [28], specifically in goal 3.4, the importance of generating strategies to promote physical and mental health throughout population is emphasised. For this reason, according to Baena-Morales and González-Víllora [29], health professionals must search for adequate strategies in order to increase population motivation and, consequently, contribute to the acquisition of SDG3 “health and wellbeing”.

Although several studies analyse profiles based on motivation [23,24,25,26], no researchers have focused on analysing profiles based on the support for perceived autonomy. The need to promote an optimum interpersonal style from the coach, is also considered here. This study centres on the support for autonomy in order to improve the motivational environment of sports practice, while taking into account the lack of bibliographical resources related to this field and the importance of promoting an interpersonal style that favours the autonomy of sportspeople. Thus, the general objective of the present study is to determine the existing profiles in the sample of athletes based on the support for perceived autonomy, with the specific objective being to analyse the differences between the profiles regarding the satisfaction of basic psychological needs, motivation, and the intention to be physically active. It is hypothesized that athletes with a high autonomy support profile will present greater satisfaction in basic psychological needs, more self-determined motivation, and a greater intention to be physically active.

## 2. Materials and Methods

### 2.1. Participants

The sample of this study was initially composed of 301 athletes belonging to the Provincial Sports Federation of Azuay in Ecuador. After excluding the questionnaires that were not entirely completed and applying the statistical procedures to detect atypical cases and missing values, the final sample comprised 280 federated athletes (male = 153 and female = 127), aged between 12 and 20 years (M = 15.28, SD = 1.71).

### 2.2. Instruments

#### 2.2.1. Support for Autonomy

The Scale of Support for Autonomy of Moreno-Murcia, Huéscar, Andrés-Fabra and Sánchez-Latorre [30] was used. The questionnaire is made up of eleven items that the participants answer about the coach’s style in practices that aim to support autonomy (e.g., “Try to make us more and more autonomous”). The previous sentence used was: “In my training, my coach …”. It consists of a Likert-type scale with five response options, from (1) Surely not to (5) Surely so. The internal consistency coefficient had a value α = 0.75.

#### 2.2.2. Basic Psychological Needs

The Psychological Need Satisfaction in Exercise Scale (PNSE) by Wilson, Rogers, Rodgers and Wild [31] was used, validated in a Spanish context by Moreno-Murcia, Marzo, Martínez and Conte [32]. The PNSE uses 18 items—six to assess each of the following needs: competence (e.g., “I feel capable of completing the most challenging exercises”), autonomy (e.g., “I think I have a voice in the exercises I do”), and relationship with others (e.g., “I think I get along well with those I interact with when we exercise together”). The previous sentence was “In my pieces of training…”, and the answers were collected on a Likert-type scale, with a score ranging between 1 (False) and 6 (True). The internal consistency revealed values of α = 0.84, α = 0.76 and α = 0.66, respectively. This last value could be considered unsuitable; however, other authors, such as Sturmey, Newton, Cowley, Bouras and Holt [33], consider values between 0.60 and 0.70 to be acceptable.

#### 2.2.3. Motivation

The questionnaire, called Behavioral Regulation in Sport Questionnaire (BRSQ) by Lonsdale, Hodge and Rose [34] and validated in Spanish by Moreno-Murcia et al. [32], was used. It consists of 36 items grouped into nine factors of four items each that measure general intrinsic motivation (e.g., “because I like it”), intrinsic motivation about knowledge (e.g., “I like to learn new things about this sport”), the intrinsic motivation of stimulation (e.g., “Because of the positive feelings I feel while practicing this sport”), the intrinsic motivation of achievement (e.g., “Because I enjoy myself while working on something important”), integrated regulation (e.g., “Because I like living according to my values”), identified regulation (e.g., “Because I appreciate the benefits of this sport”), introjected regulation (e.g., “Because I feel obliged to continue”), external regulation (e.g., “ To satisfy those who want me to play”) and amotivation (e.g., “However, I wonder why I continue”). The introductory phrase used was: “I participate in this sport …”. A seven-point Likert-type scale was used, ranging from 1 (Very false) to 7 (Very true). The reliability of the variables for Spanish athletes was α = 0.61 for general motivation; α = 0.78 for intrinsic knowledge motivation; α = 0.71 for intrinsic stimulation motivation; α = 0.75 for intrinsic achievement motivation; α = 0.77 for integrated regulation; α = 0.67 for the identified regulation; α = 0.72 for introjected regulation; α = 0.75 for external regulation; 0.70 for amotivation. In turn, the Self-Determination Index (SDI) was calculated to check the motivational orientation of the athletes. For this calculation, Vallerand [7] established the following formula: (intrinsic motivation ∗ 2) + (identified regulation + integrated regulation)/2 − (external regulation + introjected regulation)/2 − (amotivation ∗ 2).

#### 2.2.4. Future Intention to Be Physically Active

The questionnaire, called “Intention to be physically active” (IPA) by Hein, Müür and Koka [35], was used and validated in a Spanish context by Moreno, Moreno and Cervelló [36]. This questionnaire is made up of 5 items. (e.g., “I usually play sports in my spare time”). The previous sentence used is: “Regarding your intention to practice some physical-sporting activity …”. The answers are closed with a Likert-type scale whose score ranges from disagree (1) to agree (5). The reliability value is α = 0.75.

### 2.3. Procedure

The design was approved by the Research Ethics Commission of the University of Murcia, code 3023/2020. The project was presented to the Azuay Sports Federation in Cuenca, Ecuador. The data were collected from a non-probabilistic sample of athletes. Contact was made with the managers, coaches, and the coaches’ assistants of the participating federation to inform them of the objectives and request their collaboration concerning the underage athletes. A written authorization from their parents, guardians, or legal representatives was required, and once the relevant consent was obtained, the athletes could participate in the study. They were then informed of how to complete the questionnaire and resolve any doubts that could arise during this process. Then, the questionnaires were administered with the researcher present to give a brief explanation of the objective of the study. The questionnaires were administered at the beginning of training, and the anonymity of the responses was confirmed. The time required to complete the questionnaire was approximately 15 min, and it varied slightly according to the age of the athletes.

### 2.4. Statistical Analysis

First, a reliability analysis of all the scales was performed, and then the Mahalanobis distance was used to detect and eliminate those atypical cases or those that did not follow a logical pattern in the set of variables. In addition, the asymmetry and kurtosis values were analysed and considered adequate at values of <2 and <7, respectively [37], together with the Z scores. After eliminating 21 subjects who did not meet these requirements, the next step was to proceed to the reliability analysis of the different scales, finally counting on a total sample of 280 subjects. Most of the reliability coefficients revealed values above 70, a criterion considered acceptable for the psychological mastery scales [38], only three values were found in a range between 0.60 and 0.70 but were considered acceptable, following Sturmey, Newton, Cowley, Bouras and Holt [33], for the purpose of the investigation.

Second, a profile analysis was performed using the support for autonomy perceived by the coach as an independent variable. To determine the number of profiles, a cluster analysis was carried out using Ward’s hierarchical method and the most distant neighbour method, obtaining similar results. The dendrogram suggested the elaboration of two to three sets. Subsequently, the K-means method was used to make the final two clusters. Norusis [39] stated that the smaller coefficients reflect greater homogeneity among the members of a group. In addition, a double division cross-validation approach was performed to inspect stability, this the sample was randomly divided into halves, and the procedure was applied again to each subsample.

Subsequently, each profile was examined employing a MANOVA multivariate analysis, taking into account the differences found in each of the variables under investigation. A statistical analysis was performed using the IBM SSPS 24.0 package (Armonk, NY, USA).

## 3. Results

### 3.1. Descriptive Analysis and Correlations

Table 1 shows the descriptive analysis mean, standard deviation, asymmetry and kurtosis) and the reliability analysis (Cronbach’s alpha value) of the study variables. All dimensions obtained high reliability scores, except for relatedness, general intrinsic motivation and identified regulation.

Table 2 shows the significant levels and the correlation intensity (** *p* < 0.01; * *p* < 0.05) among all the variables of this study. A statistically significant correlation was shown for all of them except for autonomy with general intrinsic motivation, integrated regulation, identified regulation, external regulation, amotivation, intention to be physically active, and self-determination index. On the other hand, the need for a relationship was not correlated with intrinsic motivation towards knowledge or with external regulation, amotivation, or the self-determination index. General intrinsic motivation did not correlate with introjected regulation. The three intrinsic motivation scales did not correlate with introjected regulation. Integrated regulation did not correlate with introjected regulation and identified regulation did not correlate with external regulation. In turn, amotivation was not correlated with being physically active or with support for autonomy. Finally, the intention to be physically active was not correlated with the self-determination index or with support for autonomy.

### 3.2. Cluster Analysis

The cluster analysis was carried out according to the considerations of Hair, Anderson, Tatham and Black [40]. The obtained dendrogram suggested the existence of two clusters (Figure 1). The clusters were grouped into a “high perception of support for autonomy” (cluster 1, N = 190; 67.9%), with statistically significant higher values in the perception of autonomy and a “low perception of support for autonomy” (cluster 2, N = 90; 32.1%). The values were 4.38 (SD = 0.32) and 3.40 (SD = 0.40) in the autonomous style perception for cluster 1 and cluster 2, respectively.

### 3.3. Differences in Basic Psychological Needs, Motivation, and Intention to Be Physically Active

A multivariate analysis (MANOVA) was carried out using the clusters as independent variables and the rest of the study variables as dependent variables (see Table 3). Statistically significant differences were found at the multivariate level (Wilks’ Lamda = 0.342, F = 36.456, *p* = 0.00). The univariate ANOVAs showed statistically significant differences in most of the variables, except for introjected regulation, external regulation, amotivation, and the intention to be physically active, as shown in Table 3.

## 4. Discussion

The objective of this study was to determine the existing profiles in a sample of athletes based on the support for perceived autonomy and to analyse the differences concerning the satisfaction of basic psychological needs, motivation, and intention to be physically active.

The results obtained support the proposal that in training environments where athletes perceive high levels of support for autonomy, the satisfaction of the basic psychological needs of competence, autonomy, and relationships with others is developed, as well as the index of self-determination and the most autonomous motivation. On the contrary, athletes are frustrated in environments where they perceive that coaches provide low levels of support for autonomy, basic psychological needs, and more self-determined and autonomous motivation [3]. These outcomes are linked with the Vallerand hierarchical model of motivation [41] which assumes that some social factors are triggers and influence basic psychological need satisfaction and motivation (mediators), and these mediators, in turn, influence the cognitive, motor, social and emotional consequences.

Regarding the satisfaction of basic psychological needs, we found that the profile with a high perception of support for autonomy had a greater satisfaction of the three needs as a whole, as indicated by other studies—such as those by Álvarez et al. [19], Bartholomew et al. [16], González, Castillo, García-Merita and Balaguer [42], Gutiérrez, Sancho, Galiana and Tomás [43] and Quested et al. [44]—of federated athletes at the regional, national and international club level, with similar ages and who practiced individual and group sports. In turn, these results are also confirmed when autonomy support is separately related to each of the basic psychological needs [11,19,26,45,46,47], with athletes from different disciplines at the regional, national and international level and with a training frequency of between 3 to 5 days a week.

Support for autonomy created a positive relationship with motivation, as shown in the studies by Amorose and Anderson-Butcher [11], Gillet et al. [10] and Pelletier et al. [14]. It should even be noted that in the study by Ramis, Torregrosa, Viladrich and Cruz, [48], support for autonomy negatively predicted amotivation, coinciding with our study of athletes who received high support for perceived autonomy and had higher scores in the variables of self-determined motivation.

In the present study, however, we did not find statistically significant differences in the intention to be physically active in the future. Here, we do not corroborate the studies of Franco et al. [23], Haerens et al. [24] and Friederichs et al. [25], in schoolchildren from different countries where these variables were also analysed, as well as the study of Valero-Valenzuela et al. [26], where the intention to continue being physically active in the future was analysed in a sample of Spanish federated athletes of a single discipline, such as athletics, since they did find an existing relationship with motivation. Similarly, Almagro, Sáenz and Moreno [49], in a study of young athletes between 12 and 17 years of age in individual and collective sports and using a model of structural equations, predicted that in motivational environments granted by the coach, a greater intention to practise sports is generated. A possible explanation for the data obtained in this study could be that, since they are federated athletes, both groups started with very high scores in the intention of future practice, as indicated in the study by Franco, Tejero and Arrizabalaga [50], where it is shown that the starting levels for the intention to be physically active in federated athletes are higher than those in non-federated athletes. Therefore, this may have led to no differences between the two profiles in the present study. However, the same principle could be applied to the study by Valero-Valenzuela et al. [26], where federated athletes also participated but no differences were obtained in the motivational profiles. More studies in this regard should be carried out to clarify these results.

Considering the importance of the support of autonomy [51] that the coach provides to his athletes, it is understood that when coaches give freedom to their athletes, they take into account their opinions and feelings and offer them significant reasons to execute the training. Athletes show that their basic psychological needs are satisfied. Meaning that they feel free in their decision-making, competent in their assigned activities and sports actions, and accepted by their training partners. At the same time, it is suggested that they have more self-determined motivation to participate in their favourite sports. Quite the opposite would occur if the coach were to act in a controlling manner with the possible frustration of basic psychological needs and a reduction in a more self-determined motivation [52].

Regarding the limitations of the study, it is worth mentioning that the sample was reduced for cluster analysis, which influenced the level of significance for some variables, in addition to being chosen for accessibility and not random convenience. When analysing the respondents, sex, the type of training, training days per week, hours, and training sessions per day were taken into account, but age was not considered, which prevented the data from showing correlations with other variables as the sample was very small. Likewise, data recording has a series of limitations, since its effectiveness depends on the reading ability and the sincerity of the respondents to provide correct answers when filling out the questionnaires.

Regarding future studies, it would be feasible to carry out more profile studies based on perceived autonomy in different educational levels, such as primary and secondary school, college and university institutions, analysing the influence of a controlling style of coaching [53] or the commitment to the opposite style. Other theories that examine autonomy taking into account other variables such as group cohesion and affective states are suggested [54,55]. Similarly, in future research, it would be interesting to carry out prediction analysis based on age. Experimental studies should be considered to verify the causal effects between the variables that are analysed, and longitudinal should consider the context of sample selection.

## 5. Conclusions

In conclusion, this study explored two profiles of athletes: a profile with a perception of high support for autonomy with higher scores in the satisfaction of basic psychological needs, self-determined motivation, and the intention to be physically active. However, in this last variable, no significant differences were found concerning the group with a perception profile of low support for autonomy, which had low values in the satisfaction of basic psychological needs and in self-determined motivation. The findings are beneficial in the sense of granting autonomy support strategies to coaches to generate a more motivating environment in the athletes’ practice sessions. This is completely in line with United Nations plan with regard to SDG 3 which aims to foster “health and wellbeing” among the population.

## Figures and Tables

**Figure 1 ijerph-19-12327-f001:**
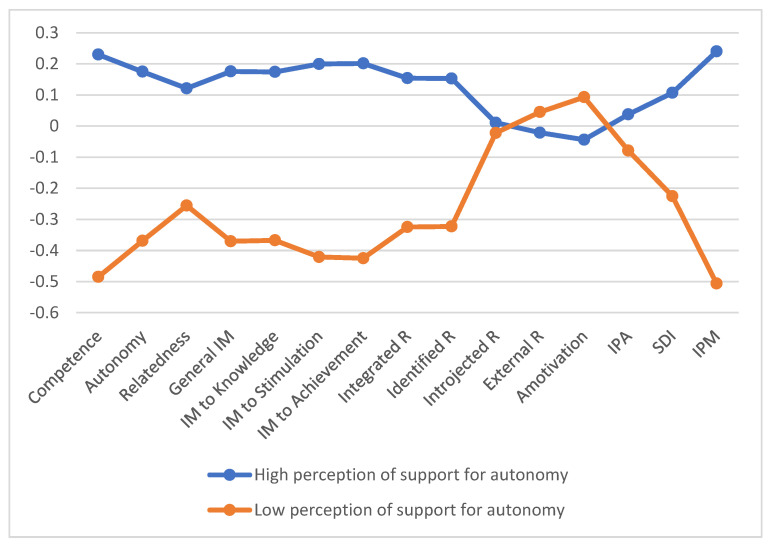
Perceived autonomy profiles for the different variables.

**Table 1 ijerph-19-12327-t001:** Descriptive analysis.

		M	SD	A	K	α
1	Competence	5.32	0.68	−1.233	1.185	0.84
2	Autonomy	3.97	1.05	−0.414	−0.176	0.76
3	Relatedness	4.95	0.79	−0.783	0.241	0.66
4	IM General	6.49	0.65	−1.651	3.305	0.61
5	IM Knowledge	6.41	0.76	−2.040	6.030	0.78
6	IM Stimulation	6.31	0.71	−1.267	1.470	0.71
7	IM Achievement	6.41	0.73	−2.004	5.108	0.75
8	Integrated R	6.23	0.89	−1.430	1.622	0.77
9	Identified R	6.33	0.74	−1.378	1.827	0.67
10	Introjected R	3.43	1.57	0.064	−0.886	0.72
11	External R	2.29	1.37	1.103	0.544	0.75
12	Amotivation	2.66	1.42	0.663	−0.142	0.70
13	IPA	4.49	0.51	−1.290	1.449	0.75
14	SDI	11.08	4.52	−0.626	−0.108	X
15	IPM	4.75	0.60	−0.490	0.078	X
16	Autonomy Support	4.75	0.60	−0.674	0.400	0.75

Note: M = mean; SD = standard deviation; A = asymmetry; K = kurtosis; α = Cronbach’s alpha value; IM = intrinsic motivation; R = regulation; IPA = intention to be physically active; SDI = Self-Determination Index; IPM = Index of Psychological Mediator.

**Table 2 ijerph-19-12327-t002:** Correlation analysis.

		2	3	4	5	6	7	8	9	10	11	12	13	14	15	16
1	Competence	0.229 **	0.402 **	0.470 **	0.400 **	0.537 **	0.558 **	0.432 **	0.395 **	−0.006	−0.139 *	−0.213 **	0.170 **	0.365 **	0.682 **	0.330 **
2	Autonomy		0.208 **	0.066	0.183 **	0.098	0.161 **	0.023	0.105	0.154 **	0.103	−0.002	0.045	−0.012	0.754 **	0.295 **
3	Relatedness			0.281 **	0.110	0.239 **	0.315 **	0.255 **	0.175 **	0.162 **	0.041	0.017	0.215 **	0.074	0.708 **	0.222 **
4	IM General				0.428 **	0.603 **	0.565 **	0.474 **	0.346 **	−0.028	−0.198 **	−0.266 **	0.280 **	0.563 **	0.336 **	0.288 **
5	IM Knowledge					0.550 **	0.487 **	0.506 **	0.538 **	0.118 *	−0.089	−0.175 **	0.166 **	0.319 **	0.303 **	0.258 **
6	IM Stimulation						0.725 **	0.573 **	0.447 **	−0.013	−0.196 **	−0.315 **	0.246 **	0.495 **	0.361 **	0.255 **
7	IM Achievement							0.607 **	0.529 **	−0.037	−0.147 *	−0.310 **	0.190 **	0.489 **	0.439 **	0.316 **
8	Integrated R								0.538 **	0.022	−0.169 **	−0.247 **	0.221 **	0.455 **	0.286 **	0.161 **
9	Identified R									0.128 *	−0.030	−0.198 **	0.147	0.341 **	0.285 **	0.265 **
10	Introjected R										0.551 **	0.455 **	0.091	−0.539 **	0.157 **	−0.017
11	External R											0.553 **	−0.011	−0.671 **	0.026	−0.025
12	Amotivation												−0.016	−0.908 **	−0.073	−0.089
13	IPA													0.110	0.184 **	0.116
14	SDI														0.162 **	0.183 **
15	IPM															0.391 **
16	Autonomy Support															

Note: ** *p* < 0.01; * *p* < 0.05; IM = intrinsic motivation; R = regulation; IPA = intention to be physically active; SDI = self-determination index; IPM = index of psychological mediator.

**Table 3 ijerph-19-12327-t003:** Multivariate analysis of basic psychological needs, motivation, intention to be physically active, and controlling style.

	High Perception of Support for Autonomy		Low Perception of Support for Autonomy			
	*M*	*SD*	*M*	*SD*	*F*	*p*
Competence	5.48	0.56	4.99	0.79	35.034	0.000 **
Autonomy	4.16	1.06	3.59	0.92	19.250	0.000 **
Relatedness	5.04	0.77	4.74	0.80	8.911	0.003 **
General IM	6.61	0.53	6.25	0.80	19.405	0.000 **
IM to Knowledge	6.55	0.66	6.14	0.88	19.062	0.000 **
IM to Stimulation	6.45	0.61	6.01	0.81	25.594	0.000 **
IM to Achievement	6.56	0.56	6.10	0.92	26.127	0.000 **
Integrated R	6.36	0.78	5.94	1.03	14.679	0.000 **
Identified R	6.44	0.69	6.09	0.79	14.493	0.000 **
Introjected R	3.44	1.66	3.39	1.38	0.066	0.797
External R	2.26	1.40	2.35	1.31	0.269	0.604
Amotivation	2.60	1.47	2.79	1.32	1.142	0.286
IPA	4.51	0.49	4.45	0.55	0.825	0.365
SDI	11.56	4.42	10.06	4.60	6.888	0.009 **
IPM	4.89	0.56	4.44	0.58	38.570	0.000 **
Wilks’s lambda (λ) = 0.342 (f = 36.456) *p* = 0.000						

Note: ** *p* < 0.01; M = mean; SD = standard deviation; IM = intrinsic motivation; R = regulation; IPA = intention to be physically active; SDI = Self-Determination Index; IPM = Index of Psychological Mediator.

## Data Availability

Data collected and analysed during the study are available upon reasonable request.

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
