# Peer review of "Contributing to SDG 3 through Support for Trainer Autonomy: Relationship with Motivation, Basic Psychological Needs and the Intention to Be Physically Active"

_ijerph, 2022, doi:10.3390/ijerph191912327_

Round 1
Reviewer 1 Report
Line 13. "For this" should be erased
- Lines 14 and 15 could be erased to have an easier abstract (From "The variables analyzed were......"
- Lines 22 and 23 are too long. Consider to split them up into short phrases.
- The expression "so" is informal, please change it. English should be revised in the entire manuscript.
- The word monitor sounds like an incorrect translation.
- When adding examples add the Latin expressions "e.g.," (line 51 and so on).
- Review the phrase in line 57.
- From lines 65 to 68 review the phrase because it is too long
- It is needed to add the novelty of the study and why to carry out this study. In other words, it is needed the rationale to create the study.
- In the participants' section please use short phrases.
- Cronbach alphas lower than .70 are consider unsuitable by some authors, please, verify this issue.
- Please review this phrase (line 154)"for written authorization from their parents"
- The word demotivation is inconsistent and sometimes changes to amotivation in the manuscript. Please, verify the correct English expression.
The discussion is too descriptive, you must explain why you think that the outcomes are.
- Limitations should be in discussion section as well as the future lines.
- Besides, it is needed to add practical implications to the discussion.
- Finally, maybe It is interesting to see other theories that examined autonomy. See for instance:
González-García, H., Martinent, G., & Nicolas, M. (2022). A Temporal Study on Coach Behavior Profiles: Relationships With Athletes Coping and Affects Within Sport Competition. Journal Of Sport &Amp; Exercise Psychology, 44(2), 94-102. doi: 10.1123/jsep.2021-0071
González-García, H., Martinent, G., & Nicolas, M. (2021). Relationships between coach's leadership, group cohesion, affective states, sport satisfaction and goal attainment in competitive settings. International Journal Of Sports Science & Coaching, 17(2), 244-253. doi:10.1177/17479541211053229
Author Response
Thank you for your comments. This is our answer:
Extensive editing of English language and style required
Line 13. "For this" should be erased
It has been changed.
- Lines 14 and 15 could be erased to have an easier abstract (From "The variables analyzed were......"
It has been changed.
- Lines 22 and 23 are too long. Consider to split them up into short phrases.
It has been fixed as you suggested.
- The expression "so" is informal, please change it. English should be revised in the entire manuscript.
It has been corrected throughout the document.
- The word monitor sounds like an incorrect translation.
It has been corrected throughout the document. (Line 41 and 153)
- When adding examples add the Latin expressions "e.g.," (line 51 and so on).
It has been corrected throughout the document. (lines 51,52,104,112,113,124,131,145,)
- Review the phrase in line 57.
It has been changed the beginning of the paragraph with “As for” for a better understanding.
- From lines 65 to 68 review the phrase because it is too long
It has been split it up into two phrases.
- It is needed to add the novelty of the study and why to carry out this study. In other words, it is needed the rationale to create the study.
The suggested recommendation has been done just before the objective adding two new sentences with the rationale to create the study (Line 79-84).
- In the participants' section please use short phrases.
It has been corrected.
- Cronbach alphas lower than .70 are consider unsuitable by some authors, please, verify this issue.
That is true. However, some authors consider values between .60 and .70 to be acceptable such as Sturmey, Newton, Cowley, Bouras and Holt (2005). This information is detailed in the Statistical analysis section. Furthermore, it has been added an extra sentence at the end of the 2.2.2. Section giving this explanation.
- Please review this phrase (line 154)"for written authorization from their parents"
It has been corrected (line 155).
- The word demotivation is inconsistent and sometimes changes to amotivation in the manuscript. Please, verify the correct English expression.
This word has been replaced by "amotivation" throughout the document.
-The discussion is too descriptive, you must explain why you think that the outcomes are.
Some new comments have been added in the discussion paragraphs, especially in those which had not any explanation, such as the second and sixth paragraphs of this section.
- Limitations should be in discussion section as well as the future lines.
It has been replaced as requested.
- Besides, it is needed to add practical implications to the discussion.
Practical implications have been added in the last paragraph before the conclusions section.
- Finally, maybe It is interesting to see other theories that examined autonomy. See for instance:
González-García, H., Martinent, G., & Nicolas, M. (2022). A Temporal Study on Coach Behavior Profiles: Relationships With Athletes Coping and Affects Within Sport Competition. Journal Of Sport &Amp; Exercise Psychology, 44(2), 94-102. doi: 10.1123/jsep.2021-0071
González-García, H., Martinent, G., & Nicolas, M. (2021). Relationships between coach's leadership, group cohesion, affective states, sport satisfaction and goal attainment in competitive settings. International Journal Of Sports Science & Coaching, 17(2), 244-253. doi:10.1177/17479541211053229
The two suggested studies have been added (line 311).
Reviewer 2 Report
Dear Authors.
It has been a pleasure to review the article. Among the variables I study related to physical activity, motivation is one of the three to which I dedicate most effort in my research.
I propose a series of changes listed below, I hope that the proposed ideas and references will add value to the work of the researchers.
Best regards
———————————————————————————-
Introduction
Line 50: In the paragraph that begins in that line I would expand on the idea of the importance and relationship of motivation (and its dimensions) with satisfaction. To this end, I propose an article that I used recently. https://doi.org/10.3390/su13063183
Line 65: Linked to the relationship between motivation and the intention to be physically active, it would be good to link it to the physical self-concept, he provided a quote and its reference as an idea.
“A better perception of one’s physical self-concept can work as a facilitator of physical activity and as a result of physical activity” https://doi.org/10.1080/02640414.2019.16413819
Line 80: reformulate the objective. I find it long and complex. I propose to formulate a general objective and one or two more specific objectives.
Material and methods
Line 91: the final sample has been composed of 280 federated athletes of 91 male (n = 153) and female (n = 127) with ages between 12 and 20 years, with the mean age 92 of (M = 15.28, SD = 1.71)
Is it more synthetic and grammatically better? (the final sample has been composed of 280 federated athletes (male = 153 and female = 127), with ages between 12 and 20 years (M = 15.28, SD = 1.71))
Line 93: The information provided there is irrelevant, the type of individual or collective sport, training ranges, etc., and then the results are not categorized or analyzed according to these differences. Therefore, I find it unnecessary.
Line 102 and 121: Inconsistency in citation, review criteria.
Results
Line 188: I would divide table 1 into two. One with the descriptive data and one that is the correlation matrix. On the one hand it is aesthetically complex and difficult to understand and on the other hand it is better to separate the two types of analysis. When writing the results of this table 1, it would be necessary to write more information about the descriptive data and to indicate the intensity of the statistically significant correlations by ranking them.
Line 211: To reposition figure 1 so that it is close and framed to point 3.2 which is where the results of the cluster analysis are explained.
Discussion
I find the discussion a bit sparse and lacking in references. I would add more information to discuss and some more references. There is a lot of published research referring to motivation and the variables studied.
Author Response
Dear reviewer, thanks for your comments. This is our answer:
Line 50: In the paragraph that begins in that line I would expand on the idea of the importance and relationship of motivation (and its dimensions) with satisfaction. To this end, I propose an article that I used recently. https://doi.org/10.3390/su13063183
What you suggested was added and the article was quoted (lines 48-50).
Line 65: Linked to the relationship between motivation and the intention to be physically active, it would be good to link it to the physical self-concept, he provided a quote and its reference as an idea.
“A better perception of one’s physical self-concept can work as a facilitator of physical activity and as a result of physical activity” https://doi.org/10.1080/02640414.2019.16413819
The idea to link the physical self-concept and physical activity has been included in the line 69 next to motivation and intention to be physically active and the paper has been quoted.
Line 80: reformulate the objective. I find it long and complex. I propose to formulate a general objective and one or two more specific objectives.
It has been included the suggestion (lines 84-91).
Material and methods
Line 91: the final sample has been composed of 280 federated athletes of 91 male (n = 153) and female (n = 127) with ages between 12 and 20 years, with the mean age 92 of (M = 15.28, SD = 1.71)
Is it more synthetic and grammatically better? (the final sample has been composed of 280 federated athletes (male = 153 and female = 127), with ages between 12 and 20 years (M = 15.28, SD = 1.71))
It has been changed (lines 97-98)
Line 93: The information provided there is irrelevant, the type of individual or collective sport, training ranges, etc., and then the results are not categorized or analyzed according to these differences. Therefore, I find it unnecessary.
As suggested it has been omitted.
Line 102 and 121: Inconsistency in citation, review criteria.
Citations have been reviewed and we consider they are consistent.
Results
Line 188: I would divide table 1 into two. One with the descriptive data and one that is the correlation matrix. On the one hand it is aesthetically complex and difficult to understand and on the other hand it is better to separate the two types of analysis. When writing the results of this table 1, it would be necessary to write more information about the descriptive data and to indicate the intensity of the statistically significant correlations by ranking them.
Table 1 was divided into two as suggested. Table 2 indicates the intensity of the statistically significant correlations by ranking them.
Line 211: To reposition figure 1 so that it is close and framed to point 3.2 which is where the results of the cluster analysis are explained.
It has been changed.
Discussion
I find the discussion a bit sparse and lacking in references. I would add more information to discuss and some more references. There is a lot of published research referring to motivation and the variables studied.
In the discussion section it has been added more information to discuss and some more references. There are some new comments in the discussion paragraphs, especially in those which had not any explanation before, such as the second and sixth paragraph of this section.
Round 2
Reviewer 1 Report
Thanks to the authors.
All the comments were addresed.
Regards